# Proteomic Study of Low-Birth-Weight Nephropathy in Rats

**DOI:** 10.3390/ijms221910294

**Published:** 2021-09-24

**Authors:** Toshiyuki Imasawa, Stéphane Claverol, Didier Lacombe, Nivea Dias Amoedo, Rodrigue Rossignol

**Affiliations:** 1Kidney Center, National Hospital Organization Chiba-Higashi National Hospital, Chiba 260-8712, Japan; 2Rare Diseases, Genetics and Metabolism, University of Bordeaux, INSERM U1211, 33000 Bordeaux, France; didier.lacombe@chu-bordeaux.fr; 3Department of Biology and Medical Sciences, University of Bordeaux, 33000 Bordeaux, France; stephane.claverol@u-bordeaux.fr; 4Functional Genomics Center, Proteomics Department, University of Bordeaux, 33000 Bordeaux, France; 5CELLOMET, CHU Pellegrin, 33300 Bordeaux, France; nivea.amoedo@u-bordeaux.fr

**Keywords:** low birth weight, nephropathy, proteomics, kidney, mitochondria

## Abstract

The hyperfiltration theory has been used to explain the mechanism of low birth weight (LBW)-related nephropathy. However, the molecular changes in the kidney proteome have not been defined in this disease, and early biomarkers are lacking. We investigated the molecular pathogenesis of LBW rats obtained by intraperitoneal injection of dexamethasone into pregnant animals. Normal-birth-weight (NBW) rats were used as controls. When the rats were four weeks old, the left kidneys were removed and used for comprehensive label-free proteomic studies. Following uninephrectomy, all rats were fed a high-salt diet until 9 weeks of age. Differences in the molecular composition of the kidney cortex were observed at the early step of LBW nephropathy pathogenesis. Untargeted quantitative proteomics showed that proteins involved in energy metabolism, such as oxidative phosphorylation (OXPHOS), the TCA cycle, and glycolysis, were specifically downregulated in the kidneys of LBW rats at four weeks. No pathological changes were detected at this early stage. Pathway analysis identified NEFL2 (NRF2) and RICTOR as potential upstream regulators. The search for biomarkers identified components of the mitochondrial respiratory chain, namely, ubiquinol-cytochrome c reductase complex subunits (UQCR7/11) and ATP5I/L, two components of mitochondrial F_1_F_O_-ATP synthase. These findings were further validated by immunohistology. At later stages of the disease process, the right kidneys revealed an increased frequency of focal segmental glomerulosclerosis lesions, interstitial fibrosis and tubular atrophy. Our findings revealed proteome changes in LBW rat kidneys and revealed a strong downregulation of specific mitochondrial respiratory chain proteins, such as UQCR7.

## 1. Introduction

Infants born at less than 2500 g are designated with a low birth weight (LBW). LBW individuals have an increased risk of heart disease, diabetes mellitus, and kidney disease. In particular, LBW is associated with an increased risk of chronic kidney disease (CKD) [1,2] or end-stage renal disease [3,4] and has become a global concern [5]. LBW is also significantly associated with a decreased number of nephrons in both humans and animals [6,7,8]. The molecular mechanism(s) underlying the deterioration of kidney function in LBW individuals remain unclear and no association with familial factors could be observed in a national registry composed of 1,852,080 individuals [9]. The current paradigm for LBW pathogenesis reflects Brenner’s theory (also known as the glomerular hyperfiltration theory), where adaptive mechanisms are activated in response to nephron loss and a subsequent increase in capillary pressure [10]. Such adaptation results in a vicious cycle promoting further progression of chronic kidney disease (CKD). Until now, the developmental origins of health and diseases (DOHaD) in the kidney have been preferentially explained by this mechanism [11,12], although the molecular pathogenetic mechanisms remain unclear [13].

The most characteristic glomerular change in LBW individuals is focal segmental glomerulosclerosis (FSGS) [14]. We previously reported that the pathological findings of human kidney biopsy specimens were similar between patients with mitochondrial DNA (mtDNA) mutations and those with LBW-related nephropathy [15]. Therefore, we hypothesized that mitochondrial defects could participate in LBW-related kidney disease. Using untargeted and powerful analytic methods as proteomics, we recently unraveled the molecular alterations in human podocytes challenged with hyperglycemia and discovered a specific program of metabolic pathway reprogramming regulated by MEF2C and MYF5 [16]. Likewise, a large-scale untargeted quantitative and comprehensive analysis of the molecular alterations in the kidney proteome could provide a dataset of shared interest for understanding the mechanisms underlying the development of LBW-related nephropathy. However, the omics of LBW kidneys remain unknown, and no dataset is available for performing pathway and biomarker analyses. In particular, the bioinformatic study of proteome changes could allow for prediction of the putative transcription factors, kinases or microRNAs involved in the observed changes based on the detailed knowledge of their targets and directionality of the expected changes in target expression. We recently used this predictive strategy to discover the role of various genetic regulators in different disease contexts [16,17,18,19]. Of note, a clearer understanding of the molecular basis of diabetic nephropathy was obtained using large-scale characterization proteomics, and innovative therapeutic approaches can be derived from this knowledge [20,21]. Therefore, similar approaches could be tested for LBW nephropathy.

To investigate molecular candidates involved in LBW-related nephropathy, we developed a rat model of human LBW-related nephropathy and performed a comprehensive untargeted proteomic study of the kidneys between LBW and NBW rats at a young age, when neither group had any pathological changes. These studies could reveal the early pathological factors underlying LBW-related nephropathy.

## 2. Results

### 2.1. Clinical and Histopathological Characteristics of LBW Rats

We generated a rat model of LBW as previously described [22] and summarized in (Figure 1A) to study early molecular alterations of the kidney. As shown in (Table 1), we compared rats in the two different body weight groups: NBW (n = 7) vs. LBW (n = 7). The body weights of LBW rats were consistently and significantly less than those of NBW rats at birth and remained lower at 4 and 9 weeks of age. The kidney weights of the LBW rats were also significantly less than those of NBW rats at 4 weeks of age. Clinical pathology examination revealed that the glomerular numbers per section of the LBW rats were significantly fewer than those of NBW rats at 4 weeks of age (Table 1). The product of glomerular number per section and kidney weight should be proportional to the total glomerular number. These products of LBW rats at 4 weeks old were significantly lower than those of NBW rats at 4 weeks old (Table 1), which should indicate that the total nephron numbers in LBW rats are lower than those in NBW rats as previously reported [6,7,8]. In addition, although glomerular sizes of LBW rats were significantly lower than those of the NBW group at 4 weeks old, the sizes became compatible between LBW rats and NBW rats at 9 weeks old (Table 1). These results suggested that the glomeruli with originally small sizes in LBW rats were enlarged with body growth, which should be due to glomerular hyperfiltration by intraglomerular hypertension. The molecular mechanisms underpinning the above-described changes in kidney physiology remain unclear and will be investigated in what follows.

### 2.2. FSGS Lesions in the Perihilar Area Were Developed in LBW Rats at 9 Weeks Old

There were no sclerotic lesions or any other pathological changes in glomeruli and tubules of either the LBW or NBW rats at 4 weeks of age (Figure 1). However, although glomeruli in 9-week-old NBW rats appeared almost normal (Figure 2a–d), FSGS with a predominance of perihilar lesions of sclerosis was apparently formed in 9-week-old LBW rats (Figure 2e–h). At 9 weeks of age, FSGS lesions were observed in 7.43% of the glomeruli in LBW rats but in only 0.48% of the NBW rats (Table 1). The incidence of interstitial fibrosis and tubular atrophy (IFTA) was also higher in LBW rats than in NBW rats at 9 weeks of age (Table 1). Furthermore, such histological damage was associated with an increase in serum creatinine levels in LBW rats (Table 1). The serum creatinine levels of the 9-week-old LBW rats were greater than those of the same age NBW rats, which indicated that the LBW rats also had worse kidney function accompanied by histological damage (Table 1).

### 2.3. Early Kidney Proteome Remodeling in LBW Rats

To unravel the early changes in LBW rats before pathological changes occurred in the kidney, we performed quantitative proteomic analyses on kidney cortices taken at 4 weeks of age, when the rats in both groups exhibited no pathological changes. A total of 1200 proteins were found in the rat kidney and 437 proteins were detected as differentially expressed (*p* < 0.05) between LBW and NBW rats at this time point (Appendix A). Further analysis of the proteomic dataset using a multiple comparison method with Benjamini–Hochberg correction allowed us to calculate the adjusted p-value (q value; Appendix A). The threshold for the false discovery rate (FDR) was set at q = 0.01. The dataset is available to the community through the public repository PRIDE using the accession number PXD018948. The volcano plot of the comparative proteomics data (using the q value) is shown in Figure 3A, and the top 10 proteins are listed in Table 2. The volcano plot showed that a large number of proteins were downregulated in the LBW kidney (top left quadrant). Several mitochondrial proteins responsible for energy transduction, such as different F_1_F_O_-ATP synthase subunits or complex III subunits, were strongly reduced in LBW kidneys (blue dots on the volcano plot), as shown in (Figure 3B). In contrast, the upregulated proteins revealed an increase in specific regulators of the cell proteome, such as Meprin A subunits α and β in the LBW kidney cortex or activator protein 1 (AP-1), among others. Lastly, several proteins belonging to the functional networks ‘cellular assembly and organization’ were identified (Appendix A), suggesting a perturbation of plasma membrane homeostasis in the LBW kidney cortex.

### 2.4. Metabolic and Signaling Pathways Altered in LBW Kidney

Analysis of the above-described proteomic data was performed using Ingenuity Pathway Analysis (IPA) software version July 2021. The LBW rat kidney differential proteome composed of 1183 proteins was compared to the IPA Knowledge database to detect enrichment in specific pathways using the ‘core analysis’ module. The results revealed that the main four pathways (ranked by log 10 *p*-value with Z > 1 and −logP < 3) altered in the LBW rat kidneys were EIF2 signaling, oxidative phosphorylation (OXPHOS; Z-score −1.9), TCA cycle, and sirtuin signaling (Figure 4A; Appendix A). Glycolysis and gluconeogenesis were also reduced, with 48% of the pathway components downregulateded in LBW kidney. These findings indicate that the main machinery required for energy transduction was inhibited in LBW kidneys, suggesting a reduction in energy metabolism. Intermediate metabolism pathways involved in catabolism and energy metabolism were also altered in the LBW kidney proteome, as shown for fatty acid oxidation (Figure 4B). A significant modulation of two pathways involved in protein translation was also observed: ‘EIF2 signaling’ (−log*p*-value = 19.8) and ‘Eif4 and P70S6K signaling’ (−log*p*-value = 11.4). Lastly, the catabolism of branched chain amino acids (BCAAs) was strongly altered in the LBW kidneys (valine degradation, −log*p*-value = 11.1).

### 2.5. Predicted Genetic Regulators Involved in LBW Kidney Remodeling

Analysis of the proteins with altered content (*p* < 0.05) in the LBW kidney allowed us to predict putative transcription factors, kinases, proteases, or even microRNAs involved in the observed changes based on the knowledge of the targets associated with specific regulators and the directionality of the expected changes. Such a bioinformatic predictive analysis was performed on the LBW kidney proteome. The results (Appendix A) identified a series of regulators potentially involved in LBW kidney proteome reprogramming, indicating that LBW kidney remodeling impacts different signaling pathways. The top two regulators (Z-score and p-value) were *rapamycin*-insensitive companion of mTOR (RICTOR) and NFE2L2 (NRF2), as a large number of their respective targets were found in the LBW differential proteome dataset (Table 3). These predictive findings unravel the complexity of the early molecules occurring in LBW kidneys and identify different signaling pathways and regulators potentially involved in the LBW kidney gene expression program and signature.

### 2.6. Mitochondrial Biomarkers of LBW Kidney Remodeling

The above-described proteomic investigation of the molecular changes occurring in LBW kidney cortices revealed the top alterations in mitochondrial proteins (Table 2). To validate these findings, we performed immunohistological analyses on kidney tissues obtained from 4-week-old LBW rats. The top downregulated protein was UQCR7, a subunit of the respiratory chain complex III (Table 2, Figure 3B). UQCR7 staining showed a positive signal in glomerular tufts, in vascular smooth muscle cells of arterioles, and along the apical membrane of distal tubular cells in NBW rats (Figure 5a–c,e–g). UQCR7 expression was significantly decreased in LBW rats compared with NBW rats (Figure 5i–k). Furthermore, proteomics revealed that key catalytic components of complex V, the F_1_F_O_-ATP synthase responsible for ATP synthesis, were also reduced in the cortex of LBW rats (Table 2, Figure 3B). Accordingly, immunohistology analysis demonstrated that ‘e’ subunit (Atp5I) expression was strongly markedly suppressed in tubular cells of LBW rats in contrast with NBW rats (Figure 5d,h,l). These findings indicate that UQCR7 and Atp5I are two mitochondrial proteins indicative of early kidney proteome alteration in LBW rats.

## 3. Discussion

In this study, we investigated the early molecular changes in the kidney proteome in a rat model of LBW-related nephropathy. We considered the hypothesis of fetal programming proposed by Brenner and colleagues [5,23] and searched for proteomic differences between the kidney cortices from rats with a low vs. normal birth weight. We first generated a rat model of LBW-related nephropathy using glucocorticoid treatment and validated several aspects of the previously described LBW-related kidney disease [6,14]. Several animal models for fetal programming of adult disease have been developed using nutrition, surgery, hypoxia, pharmacology, and stress [24]. The antenatal glucocorticoid treatment used for intrauterine growth retardation was validated in previous studies, showing a consistent reduction in glomerular number and filtration rate, increased apoptosis, and altered plasma sodium concentration [25,26,27]. However, when the rats were 9 weeks old, there was no significant difference in glomerular size between LBW rats and NBW rats. This finding suggests that glomerular hyperfiltration might occur in LBW rats and suggest the rat model used in our work is suitable, from the pathophysiology and histology standpoints, for investigating the pathogenesis of LBW-related nephropathy.

At 4 weeks of age, there were no pathological abnormalities in the kidneys of either LBW rats or NBW rats. Therefore, prospective and untargeted quantitative proteomic analyses were performed using kidney samples from 4-week-old rats to identify intrinsic factors potentially linked with the pathogenesis of LBW-related nephropathy. The proteomic analysis extensively described in our work revealed that the molecular pathways involved in energy transduction, such as oxidative phosphorylation, the TCA cycle, and glycolysis, were specifically downregulated in the kidneys of LBW rats at early time points of the disease. This observation suggests that alteration in energy metabolism is an early and intrinsic determinant of LBW kidney dysfunction. In particular, the content of respiratory chain complex III and complex V subunits was strongly reduced, as verified by immunohistology. More precisely, the expression of UQCR7, UQCR11, and Atp5I and ATP5L in tubular cells was markedly decreased in the kidneys of 4-week-old LBW rats. UQCR7 expression was markedly decreased not only in glomerular tufts but also in vascular smooth muscle cells, particularly in those of the afferent arterioles. As previously reported, the characteristic pathological change in LBW individuals is FSGS with a predominance of perihilar lesions of sclerosis, probably due to disturbance of intraglomerular hemodynamic status [14,15]. Therefore, dysfunction of OXPHOS in afferent arterioles of LBW rat glomeruli could result in autoregulation failure of intraglomerular pressure, which eventually should develop FSGS lesions. Although analyses using newborn rat kidneys are the most suitable for the pathogenesis of DOHaD in the kidney, the cortex volumes of kidneys at birth are not sufficient for molecular investigations. As we used kidneys from 4-week-old rats when no pathological abnormalities were observed, the results of proteomic and immunohistological analyses may not be the end result but could participate in the cause of nephropathy in adulthood. Our results are in agreement with previous reports that showed tubular dysfunction in LBW individuals [28,29]. The discovery of altered mitochondrial proteostasis in an LBW nephropathy rat model is in agreement with previous studies showing that mitochondria play important roles in renal pathophysiology [15,30,31,32,33,34,35,36,37]. The pathological similarities between low-birth-weight-related nephropathy and nephropathy associated with mitochondrial cytopathy were further discussed in our previous report [15].

Another finding of our study concerned the catabolism of branched chain amino acids (BCAAs), which was strongly altered in the LBW kidneys (17 proteins). Abnormal BCAA metabolism can result in BCAA depletion in the plasma, but this has not been verified in the blood of LBW patients. However, a prospective study performed on 15 patients with early stages of chronic kidney disease showed a significant decrease in plasma valine and leucine levels compared with controls [38]. BCAAs can also stimulate mitochondrial biogenesis [39], suggesting that the reduction in 97 mitochondrial proteins observed in LBW kidneys could be linked to reduced BCAA metabolism. The survey of the top ten proteomic changes revealed a significant increase in the expression of Meprin A (two subunits) in the kidney cortex of LBW rats. This protein is a protease that cleaves a large number of targets in the kidney, including extracellular matrix (ECM) proteins, modulators of inflammation, and proteins involved in the protein kinase A (PKA) and PKC signaling pathways [40]. Meprins have been implicated in the pathophysiology of diabetic nephropathy (DN), acute kidney injury (AKI), and fibrosis-associated kidney disease. Studies in diabetic mouse models suggested that Meprin A plays a protective role in the kidney [41], raising the need to evaluate in more detail the implication of Meprin A in LBW nephropathy progression or protection.

Computer analysis of the untargeted proteomic dataset generated in our work also predicted the alteration of mTOR signaling in LBW. Accordingly, a considerable number of studies have shown the involvement of mTOR and RICTOR in nephropathies [42,43]. Mitochondrial biogenesis and mTOR are two actionable pathways using pharmacological drugs such as bezafibrate and resveratrol for the former [44,45] and rapamycin or evelorimus for the latter [46]. In addition, treatments to restore kidney bioenergetics might provide therapeutic benefits. In support of this claim, recent studies using mitochondrial protective drugs showed improved renal function in swine with atherosclerotic renovascular disease, renal ischemia, and atherosclerotic renal artery stenosis [47,48,49]. Moreover, the predictive analysis of the LBW differential proteomic data identified NFE2L2 (NRF2) signaling as the main inhibited pathway. NRF2 not only protects mitochondria as a major antioxidant orchestrator but also regulates mitochondrial biogenesis [50,51]. Therefore, the reduction in respiratory chain protein levels in LBW kidneys could also be explained by the observed inhibition of NRF2. Therefore, NRF2 activators might also be interesting candidates for the prevention of LBW-related nephropathy, as considered in other nephropathies [52,53,54].

To conclude, the proteomic and histopathology study of an LBW rat model generated a unique dataset of interest for generating novel hypotheses on the pathogenesis of LBW nephropathy. We unraveled changes in proteins such as Meprin A, UQCR7/11 and Atp5I/L that occurred early in the disease process, suggesting that these elements are potential players in the molecular determinants of this disorder. However, our findings remain limited to the rat model used. Technical and methodological limitations also exist in our study, as only 1200 proteins were found in the rat kidney proteome. This number is low as compared to the reference rat proteome dataset although a recent analysis found a similar number of distinct proteins (1290) in rat kidney [55]. Therefore, our findings might have considered only the most abundant proteins of rat kidney. Our study might therefore indicate a new mechanism of DOHaD in kidney diseases other than hyperfiltration theory.

## 4. Methods

### 4.1. Rats

The rats were treated in accordance with the guidelines of the Committee on Ethical Animal Care and Use of the National Hospital Organization Chiba-Higashi National Hospital. Eight pregnant rats were fed standard chow ad libitum. By intraperitoneally injecting dexamethasone (DEXA) (0.2 mg/kg) into pregnant rats (n = 5) consecutively at 15 and 16 days of gestation, LBW rats were born at high rates, as shown in previous reports [22]. For controls, we injected the same volume of saline (n = 3). Only newborn male rats were selected for this experiment. We first measured body weights of the normal newborn male rats (n = 22), with a mean ± SD of 6.44 ± 0.47 g. We then selected seven rats weighing less than 5.51 g (under the mean—2 SD of NBW) from 38 newborn male rats obtained from DEXA-injected mothers. As normal controls, we selected seven rats weighing > 6.4 g. Consequently, the means ± SE of body weights of the rats used for the subsequent experiments were 5.01 ± 0.11 and 6.93 ± 0.10 in the LBW (n = 7) and NBW (n = 7) rat groups, respectively (*p* < 0.001, Table 1).

Newborn rats remained with their dams until they were 4 weeks old, at which point they were removed from the cages and provided standard chow ad libitum. When the rats were 4 weeks old, the left kidneys were removed. These extracted kidneys were separated for histological and proteomic analyses. The kidneys were horizontally cut, and specimens from the center were used for light microscopic analysis. For the light microscopy studies, kidney specimens were fixed in 10% neutral-buffered formalin followed by paraffin embedding and routine staining with Masson’s trichrome stain, or periodic acid–methenamine–silver (PAM) with hematoxylin and eosin (HE) (PAM-HE) stain. PAM-HE-stained images of paraffin-embedded kidney sections were imported into a vertical slide system. The glomerular size was automatically calculated by manually outlining the glomerular tuft area on the display. The kidney specimens used for the proteomics studies were only obtained from the cortex area. After separating the cortex from the medulla, the isolated cortex specimens were immediately frozen in liquid nitrogen and stored at −80 °C until further analysis. After uninephrectomy, all rats were fed a high-salt diet consisting of 8% NaCl until they were 9 weeks old. After sacrifice, the right kidneys were removed and separated for histological analyses in the same way as those at 4 weeks of age.

### 4.2. Immunohistochemical Analysis

Primary antibodies were anti-UQCR polyclonal antibody (14793-1-AP; Proteintech) and anti-ATP5I polyclonal antibody (ab122241; Abcam). Antigen retrieval was performed according to the manufacturer’s instructions. Numbers of UQCR7-positive tufts per glomerulus were counted in more than ten glomeruli per kidney section. The relative amounts of UQCR7 staining in vascular poles of glomeruli were also scored semi-quantitatively in more than ten glomeruli per kidney section as follows: 0 = negative staining, 1 = positive in less than 50% of the circumference of vascular pole, 2 = positive in equal to or more than 50% of the circumference 3 = positive in whole circumference. In addition, percentages of UQCR7-positive cells in distal tubular cells were calculated by counting more than fifty distal tubular cells per section. Positive rates of Atp5I in tubular cells were evaluated.

### 4.3. Proteomics

Sample preparation and protein digestion, nLC-MS/MS analysis, and database search and results processing were performed as described in [19].

### 4.4. Label-Free Quantitative Data Analysis

Raw LC-MS/MS data were imported into the Progenesis LC-MS 4.0 software program (Nonlinear Dynamics Ltd., Newcastle, UK). The data processing included the following steps: (i) feature detection, (ii) feature alignment across the samples, (iii) volume integration for two to six charge-state ions, (iv) normalization to the total protein abundance, (v) import of sequence information, (vi) ANOVA procedure at the peptide level, and filtering for features with values of *p*-value < 0.05, (vii) calculation of protein abundance (sum of corresponding peptide volume), and (viii) ANOVA testing with Benjamini–Hochberg correction for multiple testing at the protein level and filtering for features with values of *Adj. p* < 0.05. Of note, only nonconflicting features and unique peptides were considered for calculation at the protein level. Additionally, Progenesis performs an arcsinh correction of data before ANOVA calculation. Quantitative data were considered for proteins quantified by a minimum of two peptides. Protein abundance was normalized to the total protein content determined for each sample using in-gel densitometry methods (SDS-PAGE). MS data were also normalized in Progenesis based on feature median ratio. Furthermore, we selected proteins that showed a statistically significant > 20% change in expression levels (*Adj. p*-value < 0.05) for differences between two groups (n = 7 in each group). A global analysis of the data was performed using the computer platform Ingenuity Pathway Analysis (IPA; Qiagen). We used the ‘core analysis’ package to identify relationships, mechanisms, functions, and pathways relevant to the dataset of interest. We also used the ‘regulators’ package to identify predicted regulators of the proteomic changes.

## Figures and Tables

**Figure 1 ijms-22-10294-f001:**
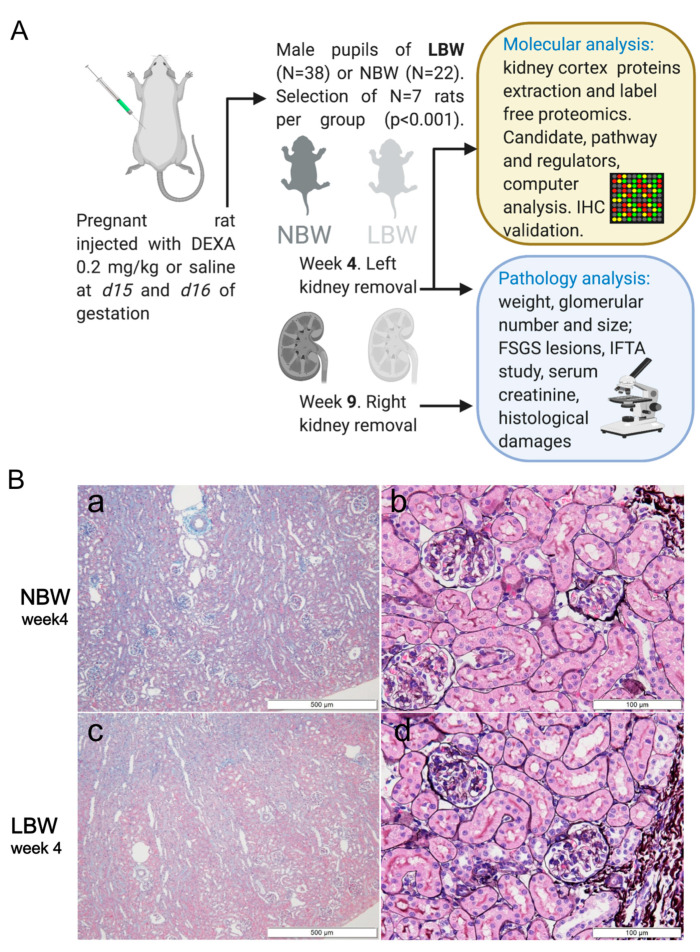
**LBW rat model generation protocol and study**. (**A**) Schematic representation of the LBW rat model generation protocol used in our study. Details on kidney sampling, clinical pathology, and molecular investigations are given in the methods. (**B**) Representative photographs of NBW (**a**,**b**) and LBW (**c**,**d**) rat kidney sections at 4 weeks of age. (**a**,**c**) are low-magnification images of Masson trichrome staining. (**b**,**d**) are photos of PAM-HE staining. Scale bars are provided in each figure. There were no pathological abnormalities in either group. Glomerular sizes in LBW rats appeared to be smaller than those in NBW rats.

**Figure 2 ijms-22-10294-f002:**
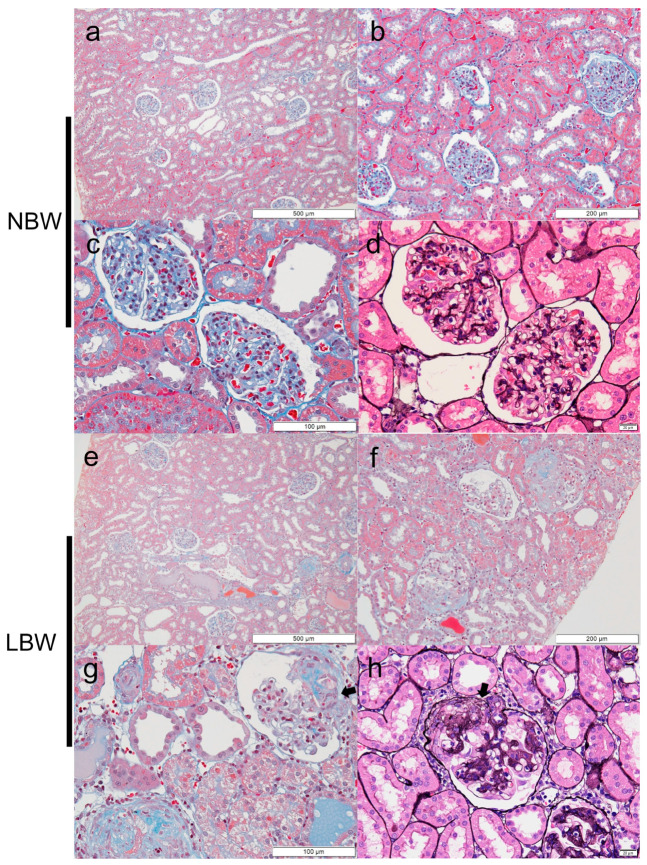
**Sclerotic lesions in LBW kidney sections at 9 weeks of age**. There were no pathological abnormalities in glomeruli of the NBW rats (**a**–**c**: Masson trichrome, **d**: PAM-HE). However, in LBW rats, sclerotic lesions were observed predominantly in the perihilar area (indicated by arrows) (**e**–**g**: Masson trichrome, **h**: PAM-HE).

**Figure 3 ijms-22-10294-f003:**
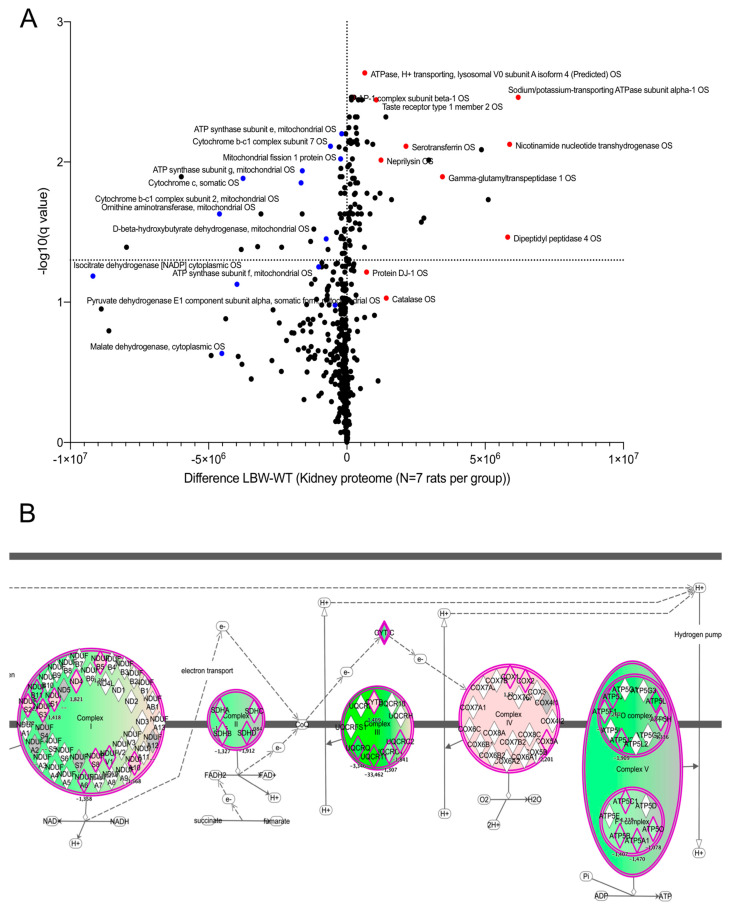
**Comprehensive label-free proteomic analysis of LBW nephropathy.** (**A**) Scatter plot (volcano plot) of the differential proteomic study between LBW and NBW kidney cortices at 4 weeks. The X-axis shows the difference in protein expression between the LBW and NBW groups (expressed as the absolute difference in protein normalized abundance). The Y-axis indicates the significance of the changes in protein content, expressed as −Log(q value). The significance threshold of *p* = 0.05 (−Log(q value) = 1.3) was selected on the graph. The reduced proteins (in the LBW group) belonging to the oxidative phosphorylation machinery are shown as blue dots on the left quadrant. (**B**) The changes in mitochondrial respiratory chain proteins are shown in a schematic representation. Each respiratory chain complex is depicted, and the subunits detected in the differential proteomic analysis are shown in green (repressed in LBW kidney) or pink (increased in LBW kidney).

**Figure 4 ijms-22-10294-f004:**
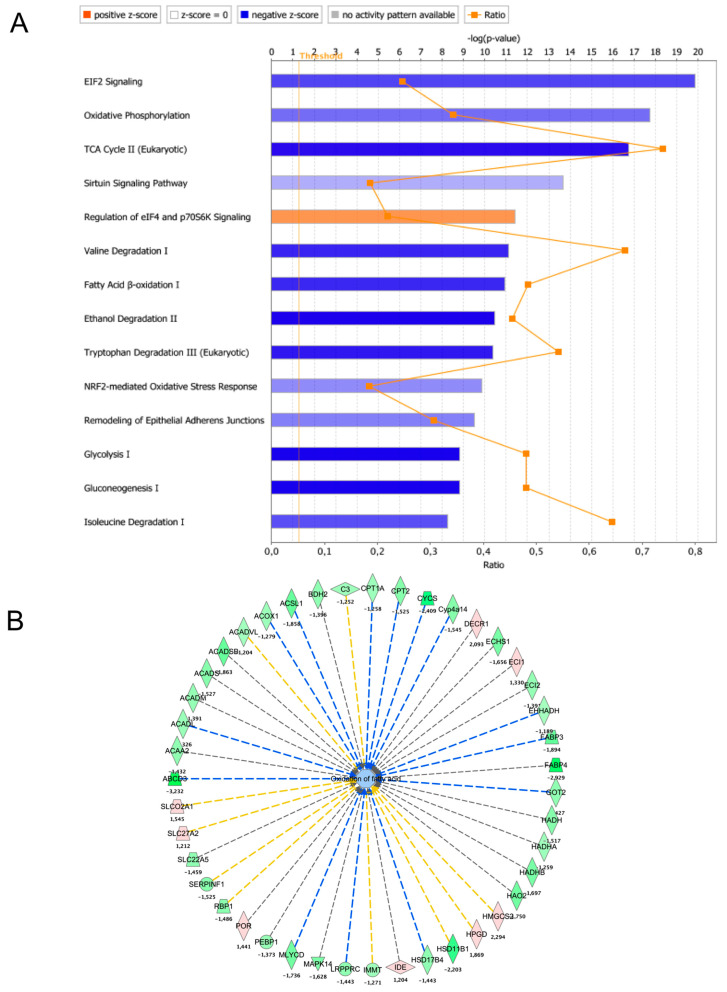
**Pathway analysis of LBW kidney proteome remodeling.** (**A**) The proteomic data were analyzed using IPA software to identify changes in signaling and metabolic pathways altered in the LBW group (core analysis; Qiagen). The background gene-set for ‘core analysis’ was set to ‘Ingenuity Knowledgebase’. The IPA canonical pathways altered in the LBW kidney were ranked by *p*-value and colored according to the Z-score (blue = pathway inhibited and orange = pathway activated). The dotted yellow lines indicate the threshold for *p*-value set at 0.05. The ratio line (orange) indicates the number of proteins altered in the dataset as compared to the number of proteins described in the IPA pathway. (**B**) Detail of the fatty acid oxidation pathway inhibited in LBW kidney. Repressed proteins are shown as green symbols, while overexpressed proteins are shown in pink. The dotted blue lines indicate a consistent observation between a predicted inhibition of the target and the actual decreased content of the protein. The lines in yellow indicate that the findings are inconsistent with the prediction. Lines in gray indicate that the effect was not predicted.

**Figure 5 ijms-22-10294-f005:**
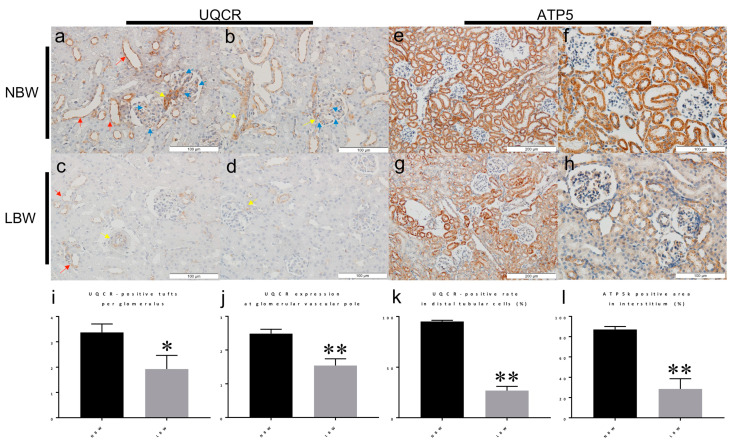
**Reduced content of UQCR and Atp5 measured by IHC in LBW kidney.** Immunohistochemistry (IHC) analyses of NBW and LBW kidney sections obtained from 4-week-old rats. UQCR7 was expressed in glomerular tufts (blue arrows), in arterioles (mainly by vascular smooth muscle cells) (yellow arrows), and along the apical membrane of distal tubules (red arrows) (**a**–**c**). On the other hand, UQCR expression was markedly decreased in LBW rats (**e**–**g**). Atp5I was positive in tubular cells in NBW rats (**d**). Atp5I expression was weakened in LBW rats (**h**). The numbers of UQCR7-positive tufts per glomerulus (**i**), semiquantitative scoring of the UQCR7 staining intensity in the glomerular vascular pole (**j**), the rates of UQCR-positive cells in distal tubular cells on their apical membrane (**k**), and the percentages of Atp5I-positive tubular cells (**l**) were statistically analyzed. * *p* < 0.05 by unpaired *t*-test. ** *p* < 0.01 by unpaired *t*-test.

**Table 1 ijms-22-10294-t001:** Basic characteristics and data from histological analysis of the rats. NBW (n = 7): group of rats born with a normal birth weight, LBW (n = 7): group of rats born with a low birth weight. Data represent mean ± SE. * The unpaired *t*-test is used for the comparison of two groups. BW, body weight; KW, kidney weight; IFTA, interstitial fibrosis and tubular atrophy. ‘na’ means not applicable.

		NBW	LBW	*p **
at birth	BW (g)	6.93 ± 0.10	5.01 ± 0.11	<0.001
at 4 weeks old	BW (g)	86.3 ± 2.6	65.0 ± 5.2	0.003
	left KW (g)	0.50 ± 0.02	0.37 ± 0.03	0.006
	left KW/BW (%)	0.58 ± 0.02	0.57 ± 0.02	0.614
	glomerular number per section	211.0 ± 5.1	186.7 ± 6.2	0.010
	glomerular number per sectionleft KW	105.2 ± 4.1	69.8 ± 7.6	0.001
	glomerular size (mm^2^)	2765 ± 65.3	2332 ± 129.8	0.011
	percentage of glomeruli with sclerosing lesions (%)	0	0	na
	IFTA (%)	0	0	na
at 9 weeks old	BW (g)	318.7 ± 10.0	266.4 ± 9.5	0.002
	glomerular size (mm^2^)	7745 ± 95.6	7557 ± 330.0	0.594
	percentage of glomeruli with sclerosing lesions (%)	0.48 ± 0.34	7.43 ± 2.15	0.008
	IFTA (%)	2.14 ± 1.01	6.43 ± 0.92	0.009
	serum creatinine (mg/dL)	0.30 ± 0.01	0.35 ± 0.02	0.028

**Table 2 ijms-22-10294-t002:** Top 10 changes in the kidney proteome in LBW rats at 4 weeks. Top ten increased and top ten decreased proteins are shown (*p* < 0.05).

Accession	ANOVA (*p*)	LBW/NBW Ratio	Description
P15800	1.54 × 10^−3^	3.03	Laminin subunit beta-2
O35077	2.59 × 10^−3^	2.87	Glycerol-3-phosphate dehydrogenase [NAD(+)], cytoplasmic
P02770	3.54 × 10^−5^	2.78	Serum albumin
D4A994	2.12 × 10^−2^	2.62	Protein Emc1
D3ZTX4	5.90 × 10^−4^	2.60	Uncharacterized protein
Q6IFW6	5.04 × 10^−3^	2.39	Keratin, type I cytoskeletal 10
Q64230	1.47 × 10^−4^	2.28	Meprin A subunit alpha
P28826	3.94 × 10^−4^	2.23	Meprin A subunit beta
G3V8X5	2.63 × 10^−3^	2.18	Protein Slc5a10
P01048	3.68 × 10^−4^	2.13	T-kininogen 1
P62898	1.41 × 10^−4^	0.42	Cytochrome c, somatic
P11517	4.97 × 10^−4^	0.41	Hemoglobin subunit beta-2
P05765	4.84 × 10^−3^	0.40	40S ribosomal protein S21
B2RYS2	8.74 × 10^−5^	0.39	Cytochrome b-c1 complex subunit 7
Q9JI66	2.10 × 10^−2^	0.36	Electrogenic sodium bicarbonate cotransporter 1
P61459	1.89 × 10^−3^	0.36	Pterin-4-alpha-carbinolamine dehydratase
Q6PDU7	2.15 × 10^−4^	0.35	ATP synthase subunit g, mitochondrial
P70623	8.68 × 10^−4^	0.34	Fatty acid-binding protein, adipocyte
A2VD16	4.75 × 10^−3^	0.32	Aldo-keto reductase family 1, member C12-like 1
P29419	1.69 × 10^−5^	0.30	ATP synthase subunit e, mitochondrial

**Table 3 ijms-22-10294-t003:** Top two regulators of the observed LBW kidney proteome reprogramming and their targets predicted by Ingenuity Pathway Analysis.

Upstream Regulator	Activation	Z-Score	*p*-Value	Target Molecules in LBW Dataset
**RICTOR**	Activated	3.842	6.28 × 10^−36^	ATP5F1A, ATP5F1B, ATP5F1C, ATP5MF, ATP5MG, ATP5PB, ATP5PD, ATP5PF, ATP5PO, ATP6AP1, ATP6V0A4, ATP6V0D1, ATP6V1A, ATP6V1B2, ATP6V1C1, ATP6V1C2, ATP6V1E1, ATP6V1F, BSG, COX5A, NDUFA10, NDUFA6, NDUFB5, NDUFS1, NDUFS2, NDUFS3, NDUFS8, NDUFV1, PPA2, PSMA1, PSMA5, PSMA6, PSMB1, PSMB3, PSMB5, PSMB7, PSMD1, PSMD11, PSMD12, PSMD2, PSMD3, PSMD6, PSMD7, PSME2, RPL10, RPL10A, RPL13A, RPL14, RPL17, RPL23, RPL26, Rpl34 (includes others), RPL4, RPL6, RPL7A, RPS10, RPS11, RPS18, RPS19, RPS2, RPS21, RPS24, RPS29, RPS3, Rps3a1, RPS6, RPS8, SDHA, SDHB, SDHC, SDHD, Uba52, UQCR7, UQCRC1, UQCRC2, UQCRQ
**NFE2L2**	Inhibited	−4.341	7.29 × 10^−36^	AKR1A1, AKR1B10, AKR7A2, ALDOA, ARF1, ATP1A1, CALB1, CAT, CCT3, CCT7, CTSD, Cyp4a14, CYP4A22, DAD1, DDX39B, DNAJB11, DNAJC3, EIF2S1, EIF3E, EIF3G, ENTPD5, EPHX1, ERP29, ESD, FABP4, FMO1, FN1, FTL, G6PD, GCLC, GCLM, GNAI2, GPX1, GSR, GSS, Gsta1, GSTA3, GSTA5, GSTM1, GSTP1, HPRT1, HSP90AA1, HSP90AB1, HSP90B1, HSPA9, HYOU1, IDE, IDH1, INMT, LMNA, ME1, MGLL, NCKAP1, OAT, PAFAH1B1, PDIA3, PDIA4, PDIA6, PFN2, PGD, PPIB, PRDX1, PSAT1, PSMA1, PSMA5, PSMA6, PSMB1, PSMB3, PSMB5, PSMD1, PSMD11, PSMD12, PSMD3, PSMD7, RACK1, RAN, RARS, RPS16, RYR3, SEC23A, SLC16A2, SLC3A1, SLC7A8, SOD1, SOD2, TALDO1, TCN2, TKT, Tpm1, TTR, TUB, TXNRD1, UGDH, UGT1A1, VCP

## Data Availability

The proteomic dataset is available to the community through the public repository PRIDE using the accession number PXD018948.

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
