# Peer review of "Proteomic Study of Low-Birth-Weight Nephropathy in Rats"

_ijms, 2021, doi:10.3390/ijms221910294_

Round 1
Reviewer 1 Report
The paper investigates the the role of molecular pathogenesis and biomarkers of low-birth-weight nephropathy in rats. That is an interesting topic and article is well written.
However, the following questions and concerns remained unanswered.
Comment 1: The title is not appropriate and should be rewritten (like »Immunohistochemical and proteomic study…« or »Immunohistochemical assessment and ...” Namely, clinical pathology is very broad term mainly concerning laboratory analysis of bodily fluids, and tissue homogenates or extracts using the tools of chemistry, microbiology, hematology and molecular pathology, but not patohistological assessment nor IHC.
Comment 2: Remember to define the non-standard abbreviations before first use, e.g. IHK, especially in the abstract.
Comment 3: Please consider editing the manuscript (font and size in the abstract).
Comment 4: Figure 5. a,b,c,d IHK staining is very pale and misleading. UQCR7 staining in glomerular tuft and along the apical membrane of distal tubules is not seen.
Comment 5: line 28: focal and segmental glomerulosclerosis
Author Response
Dear reviewer #1, thank you for your evaluation of our study. We provide below answers to your comments:
Comment 1: The title is not appropriate and should be rewritten (like »Immunohistochemical and proteomic study…« or »Immunohistochemical assessment and ...” Namely, clinical pathology is very broad term mainly concerning laboratory analysis of bodily fluids, and tissue homogenates or extracts using the tools of chemistry, microbiology, hematology and molecular pathology, but not patohistological assessment nor IHC.
We have changed the title to: Proteomic study of low-birth-weight nephropathy in rats
Comment 2: Remember to define the non-standard abbreviations before first use, e.g. IHK, especially in the abstract.
We have now defined the non-standard abbreviations in the abstract as LBW
Comment 3: Please consider editing the manuscript (font and size in the abstract).
The abstract was edited by American Journal Experts and the IJMS team.
Comment 4: Figure 5. a,b,c,d IHK staining is very pale and misleading. UQCR7 staining in glomerular tuft and along the apical membrane of distal tubules is not seen.
We have now replaced the figure with high-resolution images with arrows to indicate the positive areas.
Comment 5: line 28: focal and segmental glomerulosclerosis
We corrected the mistake
Reviewer 2 Report
The manuscript by Imasawa et al. combines a proteomic analysis and an immunohistochemical study of the kidney cortex from rats presenting with low birth weight vs. normal birth weight, the former corresponding to an established model based on dexamethasone treatment of pregnant animals. Since differential proteomics was performed at 4 weeks and clinical pathology at 9 weeks, protein expression changes are considered as early events eventually leading to detectable tissue lesions. The results reveal an impairment in metabolic processes mostly related to mitochondrial dysfunction.
Comments:
- In general ANOVA tests are used when comparing more than two groups. In this case there are only two groups. Why was ANOVA chosen and not T-test?
- On table S1, the Q-values allow to separate true from false discovery proteins (if I understood correctly). There is a “discovery” column where “yes” or “no” is indicated. I guess this obeys to q-values being over or under a threshold, and this threshold seems to be 0.01. This should be explained in the legend of table S1, and briefly indicated in paragraph 2.3 of the results.
- On Table 1, some sclerosing lesions are reported in the NBW control group of animals (0.48%). Although this is very infrequent, do the authors have any explanation for this finding? Was this expected?
- In figure 4 “mitochondrial dysfunction”, which the most significant canonical pathway in the study, does not appear.
- The legend of figure 4 should be improved. In 4A, what is the meaning of the yellow threshold line? What does the blue color scale stand for? What is the meaning of the “ratio” line? In 4B it should be explained the color code (gray, blue, yellow) of the spokes.
Author Response
thank reviewer #2 for his comments and we provide below a point-by-point answer.
2.1 In general ANOVA tests are used when comparing more than two groups. In this case there are only two groups. Why was ANOVA chosen and not T-test?
We agree. The proteomics facility used the Progenesis software for the analysis of the raw mass spectrometry spectra and this software did not allow to perform a t-test (only Anova is proposed). However, when comparing two groups with equal variances the Anova test can be used as the t test. This is why we could use Anova.
2.2 On table S1, the Q-values allow to separate true from false discovery proteins (if I understood correctly). There is a “discovery” column where “yes” or “no” is indicated. I guess this obeys to q-values being over or under a threshold, and this threshold seems to be 0.01. This should be explained in the legend of table S1, and briefly indicated in paragraph 2.3 of the results.
We have now added these precisions in the legend of Table S1 and in paragraph 2.3 (The threshold for the false discovery rate was set at q value = 0.01. )
2#3 On Table 1, some sclerosing lesions are reported in the NBW control group of animals (0.48%). Although this is very infrequent, do the authors have any explanation for this finding? Was this expected?
At 9 weeks of age, FSGS lesions were observed in 0.48% of the NBW rats presumably because of glomerular hyperfiltration after uninephrectomy (Table 1). On the other hand, such FSGS lesions were significantly increased to 7.43% in the glomeruli of LBW rats at 9 weeks of age (Table 1).
2#4 In figure 4 “mitochondrial dysfunction”, which the most significant canonical pathway in the study, does not appear.
Several Qiagen Ingenuity pathways can describe mitochondrial metabolism and there is redundancy. Each pathway contains a common set of protein with additional specific ones. In Figure 4, the pathways 'oxidative phosphorylation', 'sirtuin signaling' and 'TCA cycle' were inhibited (blue bars) and refer to mitochondrial metabolism.
2#5. The legend of figure 4 should be improved. In 4A, what is the meaning of the yellow threshold line? What does the blue color scale stand for? What is the meaning of the “ratio” line? In 4B it should be explained the color code (gray, blue, yellow) of the spokes.
We added the missing informations as follow:
The IPA canonical pathways altered in the LBW kidney were ranked by p value and colored according to the Z-score (blue = pathway inhibited and orange = pathway activated). The dotted yellow line indicate the threshold for p value set at 0.05. The ratio line (orange) indicates the number of proteins altered in the dataset as compared to the number of proteins described in the IPA pathway. (B) Detail of the fatty acid oxidation pathway inhibited in LBW kidney. Repressed proteins are shown as green symbols, while overexpressed proteins are shown in pink. The dotted blue lines indicate a consistent observation between a predicted inhibition of the target and the actual decreased content of the protein. The lines in yellow indicate that the findings are inconsistent with the prediction. Lines in gray indicate that the effect was not predicted.
Round 2
Reviewer 1 Report
The authors answered all questions.